# Anxiety, Optimism, and COVID-19 Vaccine Hesitancy among Students in a University in Southern Thailand during the 2021 Academic Year

**DOI:** 10.3390/vaccines11071157

**Published:** 2023-06-26

**Authors:** Patcharawan Kaewkrajang, Chonnakarn Jatchavala, Tharntip Sangsuwan

**Affiliations:** 1Department of Family and Preventive Medicine, Faculty of Medicine, Prince of Songkla University, Songkhla 90110, Thailand; patcharawan.kaew@gmail.com; 2Department of Psychiatry, Faculty of Medicine, Prince of Songkla University, Songkla 90110, Thailand; jchonnak@gmail.com

**Keywords:** anxiety, COVID-19, vaccine hesitancy, optimism, student

## Abstract

This study examined the prevalence of COVID-19 vaccine hesitancy and its associated factors, including anxiety and optimism, surrounding the COVID-19 pandemic among Thai university students. An online observational cross-sectional study was conducted between November and December 2021. Descriptive analyses and logistic regression were performed using R. In total, 409 university students participated in the study. Most reported high anxiety levels (69.4%), while more than half also expressed high levels of optimism associated with the COVID-19 pandemic (51.1%). Only 4.4% were unvaccinated, among whom 50% were hesitant to receive vaccination (2.2%). Per statistical analysis, those who perceived the available vaccines as low safety were significantly associated with a higher risk of vaccine hesitancy (*p* = 0.03). In contrast, those who reported that they would likely to receive the COVID-19 vaccine if recommended to by their doctors or pharmacists were significantly associated with lower vaccine hesitancy (*p* = 0.048). However, both anxiety and optimism regarding the COVID-19 pandemic were not associated with vaccine hesitancy. Thus, healthcare workers play an important role in COVID-19 vaccination counseling to prevent transmission. Health education during the pandemic should focus on COVID-19 infection as well as COVID-19 vaccines, including their safety and their adverse effects.

## 1. Introduction

The COVID-19 disease is caused by the SARS-CoV-2 virus and was first identified in late December 2019, consequently resulting in the ongoing worldwide pandemic. Patients infected with COVID-19 can exhibit respiratory symptoms as well as symptoms related to other systems including dysgeusia, anosmia, skin lesions, and gastrointestinal symptoms. In severe cases, patients can experience complications such as pneumonia, acute kidney injury, or even death [1].

While COVID-19 is self-limiting among most infected individuals, the disease is still associated with a serious loss of life worldwide. COVID-19 prevention measures include wearing a mask, handwashing, social distancing, and quarantining. These measures have significantly affected people’s daily living as well as the economy [1]. Consequently, COVID-19 vaccines have been developed, increasing optimism for prevention as well as the reduction of transmission and severe disease progression [1]. While no vaccine is 100% effective, when used broadly in communities, several vaccine preventable diseases could be eliminated and some may be eradicated. High vaccine uptake rates, specific to each vaccine preventable disease, are needed for community-level immunity to be achieved and sustained [2]. However, a vaccine is effective only for people who are vaccinated. Vaccine hesitancy refers to a delay in vaccine acceptance despite the global availability of vaccine services. Thus, vaccine hesitancy presents a complex and contextually specific condition, as it varies across time, location, and vaccine distribution [2]. In 2019, the World Health Organization listed vaccine hesitancy among the top 10 threats to global health [3].

In Thailand, 10.4% of the general population reported hesitancy approximately 6 months before the current study commenced [4]. Furthermore, the prevalence of vaccine hesitancy among senior outpatients was reported to be 44.3% during the same period [5]. At the time of this study, the COVID-19 vaccination coverage in Thailand, for one dose and two doses, were approximately 60% and 44%; respectively (as of 1 November 2021), at which time individuals working in healthcare, people with chronic disease, and the elderly were of the highest priority: university-aged students were a low priority group. However, university-aged students comprise a unique demographic group with various underlying illness experiences compounded by media and information consumption, in contrast to older groups in the population [6]. Moreover, university students might not have pre-existing medical conditions that present a high risk for severe COVID-19-related illness, but they are at risk of becoming infected and being a source of transmission to high-risk populations [7]. A study in France demonstrated that 17% of students decided against receiving a vaccination, while 25% were unsure. In particular, female students studying both pure and applied sciences were significantly associated with a higher risk of vaccine hesitancy [7]. Moreover, a study investigating Thai undergraduate students revealed that 40% of the participants reported vaccine hesitancy during the peak of the third pandemic wave in 2021 [8]. Thus, it is crucial to understand these young adults and explore their vaccination intention over time to help inform the development of the public health policy [6].

Several studies conducted at the beginning of the COVID-19 pandemic indicated that Europeans were overly optimistic regarding the infection [9,10]. Research suggests that the unrealistic optimism is related to lower preventive healthcare and behaviors [11]. Simultaneously, anxiety surrounding COVID-19 infection among Northern Thai citizens significantly influenced an increase in health-promoting and social behaviors, such as consuming vitamins, during the pandemic [12]. Anxiety is the anticipation of future threat; and is additionally often associated with vigilance in preparation for future danger [13]. Therefore, this study tested the hypothesis that optimism is related to a higher level of vaccine hesitancy, while a higher level of anxiety regarding COVID-19 infection is associated with a higher rate of vaccine acceptance.

The main objective of this study was to explore the levels of COVID-19 vaccine hesitancy, optimism, and anxiety surrounding COVID-19 among university students in Southern Thailand. Furthermore, the current study investigated the factors associated with vaccine hesitancy among students enrolled during the 2021 academic year.

## 2. Materials and Methods

### 2.1. Study Design and Setting

The current study used an observational cross-sectional design and was conducted among students from the Prince of Songkla University, the Hatyai campus, and the Pattani campus, located in the restive area of the south Thailand insurgency. The data were collected between November and December 2021. The questionnaires were completed using a Google form electronically distributed to the students through a mailing list and the group in LINE application. The participants were required to complete the anonymous online questionnaire upon providing informed consent using E-consent. This study was approved by the Human Research Ethics Committee, Faculty of Medicine, Prince of Songkla University (Ref no: REC.64-492-9-4).

### 2.2. Study Sample, Sampling, and Sample Size Calculation

The participants were Thai students, aged ≥18 years, who studied at Prince of Songkla University: Hatyai and Pattani campuses. Students diagnosed with psychiatric conditions were excluded from the study. The sample size was determined using Cochran’s formula with convenience sampling. The sample size was calculated by using data from a previous study by Tavolacci et al. [7]. The proportion of COVID-19 vaccine hesitancy in French university students was 0.42, with a confidence level of 95% and error margin of 5%. Thus, a minimum of 375 participants was required for this study.

### 2.3. Measurements

The questionnaires were developed based on a literature review. Three experts validated the questionnaires to demonstrate an Item Objective Congruence (IOC) range of 0.67–1.00. A pilot study was conducted with 30 students to measure internal consistency, which produced a Cronbach’s alpha of 0.72. The questionnaire was divided into the following five sections:(1)Demographic data, including age, gender, course of study, hometown, household income, religion, and underlying disease.(2)Perception and personal experience of both the COVID-19 pandemic and the COVID-19 vaccine; including those likely to receive the COVID-19 vaccine if recommended to by their doctors or pharmacist (answer: Yes/No), have history of COVID-19 infection (answer: Yes/No), have friends/relatives previously infected with COVID-19 (answer: Yes/No); perceptions of COVID-19’s severity, perceptions of the safety of COVID-19 vaccines and perceptions of the effectiveness of COVID-19 vaccines (each perception answer: None/Low/Moderate/High/Very high).(3)Vaccination status and willingness to receive the vaccine, whereby the respondents could indicate their willingness as “Yes”, “No”, or “Not sure”, after which respondents were directed to checklists for either a “Yes” response or a “No” or “Not sure” response to justify the reasons for their answers [7]. The respondents can choose more than one reason.(4)COVID-19-outbreak-specific optimistic explanatory style scale. A 15-item questionnaire in the Thai language, designed to measure individuals’ level of optimism during COVID-19. This uses a 5-point Likert scale; wherein: 5 represented “strongly agree” and 1 represented “strongly disagree.” The levels of optimism were classified into 5 groups (very low, low, moderate, high, and very high) (α = 0.83) [14].(5)COVID-19 anxiety scale, a 9-item questionnaire in the Thai language, designed to measure the level of anxiety related to the COVID-19 virus. This uses a 10-point Likert scale; wherein: 10 represented “most anxious” and 1 represented “least anxious.” The levels of anxiety were classified into 3 groups (low, moderate, and high) (α = 0.81) [15].

### 2.4. Statistical Analysis

A descriptive analysis was performed to define the distribution of the demographic characteristics. All variables were considered categorical variables and described as percentages based on a 95% confidence interval. Subsequently, Fisher’s exact test was used. Intention to vaccinate against COVID-19 was identified as the dependent variable for the logistic regression model. Vaccine hesitancy included both those who responded “No” and those who responded “Not sure” to the COVID-19 vaccine intention questions. The vaccine acceptant included both those who responded “Yes” to the COVID-19 vaccine intention question and those who had already been vaccinated. Data analyses were conducted using the R software package (version 4.1.3), and *p*-values of <0.05 were considered statistically significant.

## 3. Results

### 3.1. Demographic Data, Optimism, Anxiety Regarding the COVID-19 Pandemic, and Vaccination Perceptions among Southern Thai University Students

This study comprised 409 students, most of whom were female (77.8%), healthy (94.9%), Buddhist (61.6%), and aged between 18 and 23 years (85.8%). Most of the participants exhibited high levels of optimism regarding the pandemic (51.1%). Furthermore, most of the participants lived in Southern Thailand (93.7%), of which 33.9% were in the restive area of the south Thailand insurgency. Their monthly household income ranged mostly from 15,000 to 50,000 baht (49.4%). While 7.3% of the participants had been infected with COVID-19, 52.3% reported that their friends or relatives had already been infected with COVID-19. Most of the students perceived the severity of the COVID-19 pandemic to be high (51.1%) during the period of the survey. Moreover, 69.4% of the students reported high levels of anxiety concerning the COVID-19 infection. Furthermore, most of the participants exhibited high levels of optimism regarding the pandemic (51.1%). Moreover, they indicated that their decision to vaccinate would be informed by their medical doctor’s or pharmacist’s recommendation (89.0%). There were statistically significant differences by perceptions of vaccine safety, perceptions of COVID-19’s severity, household income, and recommendations by personal medical doctors and pharmacists (*p* < 0.05). Table 1 exhibits the demographic characteristics of the participants.

### 3.2. COVID-19 Vaccination Status and Willingness among Southern Thai University Students

Most of the students had already received at least two doses of the vaccination (88.3%). Furthermore, 7.3% of the participants had received one dosage, while only 4.4% of the participants were unvaccinated. Among the unvaccinated participants, 50% were willing to receive a COVID-19 vaccine in the future (2.2%). Overall, the prevalence of vaccine hesitancy among university students studying in Southern Thailand was 2.2% (Table 2).

Table 3 illustrates that the main motivation for the vaccinated individuals was to swiftly return to normal life (89.3%). Other motivations to be vaccinated were that they did not want to be infected with COVID-19 or transmit it to others (73.0% and 70.5%, respectively). The participants who reported vaccine hesitancy primarily lacked trust in the efficacy of the COVID-19 vaccines available in Thailand (88.9%). Other reasons that were provided indicated that the participants had considerable distrust in the Thai public authorities who were responsible for COVID-19 control (77.8%). Furthermore, they expressed concern regarding the adverse effects of COVID-19 vaccines (77.8%; Table 4).

### 3.3. Factors Associated with Vaccine Hesitancy among Southern Thai University Students

Table 5 shows that the university students who perceived the COVID-19 vaccine to be unsafe or of low safety were approximately 43.94 times more likely to demonstrate vaccine hesitancy than individuals who did not report such perceptions (95% CI = 3.61, 533.57). Moreover, the participants who reported that they would likely receive the COVID-19 vaccine if recommended to by their doctors or pharmacists were significantly associated with having a 0.16 times lower risk of vaccine hesitancy (95% CI = 0.03, 0.99).

Despite these findings, the level of optimism and anxiety regarding the COVID-19 pandemic were not associated with vaccine hesitancy according to the statistical analyses.

## 4. Discussion

This is the first study to focus on a younger population receiving higher education in Southern Thailand, and represents diverse religions, ethnicities, and cultures across the country. Violent situations occurring during the Southern Thailand insurgency in the southernmost border provinces, significantly disturbed public mental and physical healthcare services [16]. Previous studies of vaccine hesitancy focused on fear and anxiety toward the COVID-19 pandemic. However, this study focused on whether an optimistic attitude toward the pandemic influences vaccine hesitancy. Based on current research, this study is the first to evaluate the association between vaccine hesitancy, anxiety, and optimism regarding the COVID-19 pandemic. According to the WHO, the COVID-19 pandemic appears to be in transition, and there are risks of emergence of new variants and future surges [17]. Thus, it is essential to investigate vaccine hesitancy to develop delivery approaches for vaccination among various populations, so as to develop a robust public health policy and to enhance control of the pandemic.

The prevalence of vaccine hesitancy in this study was 2.2%. This is the lowest prevalence reported in Thailand, even during the most recent period of the survey. Another study, investigating Thai university students, revealed that 40% of the participants reported vaccine hesitancy during the peak of the third pandemic wave in 2021 [8]. Furthermore, the current findings revealed a prevalence much lower than in studies conducted in other countries; such as a survey of Chinese, French, and Canadian university students (23.9%, 25.0%, and 19.9%, respectively) [6,7,18]. However, the varying prevalence of vaccine hesitancy among university students may be related to when the surveys were conducted and the various questionnaires used [19]. Consistent with the results of this survey, the perception of the severity of the pandemic may also differ based on the period and areas in which the studies were conducted. The participants that reported vaccine hesitancy primarily lacked trust in the efficacy of the COVID-19 vaccines available in Thailand. A study in Thailand investigating vaccine hesitancy and determinants among physicians in a university-based teaching hospital in Thailand reported that vaccine hesitancy was associated with a preference for specific vaccines; especially for mRNA vaccines [20], while only CoronaVac and Sinopharm, an inactivated vaccine and AstraZeneca/Oxford, a viral vector vaccine, were available for university-aged students in Thailand at the time of the survey. Another reason, mostly cited in concerns of the vaccine hesitant, was a distrust in the Thai public authorities who were responsible for COVID-19 control. A study in Nigeria found that distrust in government was strongly associated with COVID-19 vaccine hesitancy [21]. Therefore, policy makers should be cautious when it comes to strategizing for COVID-19 vaccine distribution, especially in places where trust in government is weak [21].

According to the current survey, 95.6% of southern Thai university students had already received a vaccination. Their main reason for vaccination was to return to normal life as soon as possible and to prevent infection. This result demonstrates that the university students were aware that vaccination is important for returning to “normal life”. Furthermore, the current findings indicated that approximately 70% of students reported high levels of anxiety regarding the COVID-19 pandemic. However, the statistical analysis indicated that this was not significantly associated with vaccine hesitancy. An interesting issue is how “abnormal life” impacted their mental health and socioeconomics, as the students with household incomes below 15,000 baht per month expressed more vaccine hesitancy. A study in Thailand investigated willingness to pay for the COVID-19 vaccine, which was reported to cost approximately between 501 and 1000 baht [4], while some vaccines cost 3800 baht per two doses at that time [22]. Notably, this has presented a financial burden for students during the pandemic, when the global economy has declined [23]. Therefore, COVID-19 vaccines that have received safety approval from the Food and Drug Administration should be prioritized as an essential public health service. In particular, the COVID-19 vaccination service should be free of charge and convenient to access [24]. Other motivations to be vaccinated were that they did not want to be infected with COVID-19 or transmit it to others (73.0% and 70.5%, respectively). This result demonstrates that the students not only think of their own health benefits, they also think of others. The vaccination information emphasizing self-benefits, or the benefits of others, could increase their willingness to receive a COVID-19 vaccine [25]. Furthermore, another study has indicated that altruism is associated with the high intention to receive a COVID-19 vaccine [26]. Thus, promoting altruism could effectively motivate vaccination against COVID-19.

The current findings suggested that most of the respondents would likely receive the COVID-19 vaccine if recommended to by their doctors or pharmacists (89.0%). These individuals were significantly associated with a lower risk of vaccine hesitancy. Similarly, this finding is consistent with a review of vaccine coverage indicating that healthcare workers play an important role in promoting COVID-19 vaccination among individuals pursuing higher education [6]. A study of COVID-19 vaccine hesitancy among different population groups in China also reported that recommendations from physicians functioned as a motivator for participants to get vaccinated, and people would get the COVID-19 vaccine if their physicians recommend it [18]. Hence, vaccine promotion campaigns may need to emphasize the importance of talking with a health care provider about vaccines; including asking for information to address any concerns or questions. At the same time, health care providers should be kept informed regarding COVID-19 vaccine information and be prepared to advise any clients [26].

Another factor found to be associated with vaccine hesitancy was the perception of COVID-19 vaccine safety. According to this survey, the university students who perceived the COVID-19 vaccine to be unsafe, or of low safety, were more likely to demonstrate vaccine hesitancy than individuals who did not report such perceptions. This finding is consistent with a study from France [7]. According to the Working Group Determinants of Vaccine Hesitancy Matrix, an individual’s perception of vaccine safety is a strong predictor of hesitancy [2]. In addition to COVID-19 infection, the current results substantially suggest that health education during the pandemic should also focus on ensuring that information on COVID-19 vaccines is accessible, including vaccine safety and associated adverse effects. This study can be applied to the Thai government policy of young people’s COVID-19 vaccine acceptance.

In this study, women were not associated with vaccine hesitancy, whereas this was found to be a factor in another study from Thailand [4]. Moreover, this study found no difference in vaccine hesitancy between the students from the hometowns located in the armed conflict areas of the Southern Thailand insurgency, and those from hometowns in other provinces.

This study had several limitations; as it is a cross-sectional study design, the result might not be able to establish long term trends of vaccine hesitancy. The low rate of response might be due to using the online survey; additionally, the survey was distributed during the beginning of the semester because, due to COVID-19 situation, the students had to study online, not at the university. Thus, some students might not have had convenient access to the questionnaire. Furthermore, this study used convenience sampling; wherein, despite surveying students at both campuses located in the south of Thailand, this might have influenced generalizability among the Thai university students. Future studies should also consider the possibilities of SARS-CoV-2 virus mutation as well as the fact that the vaccines offered in Thailand could be ineffective. Alternatively, the COVID-19 pandemic could resolve and become an endemic disease similar to previous viral diseases. Moreover, a cohort study design using a qualitative method should be applied to understand the paradigm of vaccine hesitancy. In addition to COVID-19 vaccination, this study may be beneficial for improving health policies related to other vaccines to prevent inevitable infectious diseases.

## 5. Conclusions

Vaccine hesitancy among university students in Southern Thailand had a prevalence of merely 2.2%. Furthermore, 88.3% of the students were fully vaccinated in the academic year of 2021. Notably, most of the students expressed high levels of both anxiety and optimism regarding the COVID-19 pandemic. However, these were not significantly associated with vaccine hesitancy based on statistical analysis. The statistically significant associated factors were perceptions of vaccine safety and encouragement by personal medical doctors and pharmacists. Thus, healthcare workers play an important role in COVID-19 vaccination counseling to prevent transmission. Health education during the pandemic should focus on COVID-19 infection as well as COVID-19 vaccines, including their safety and adverse effects.

## Figures and Tables

**Table 1 vaccines-11-01157-t001:** Demographic variables of university students in Southern Thailand.

Variables	COVID-19 Vaccine Hesitancy N = 9 (%)	COVID-19 Vaccine Acceptance N = 400 (%)	Total N = 409	*p*-Value Fisher’s Exact Test
Age				1
18–23	9 (100)	342 (85.5)	351 (85.8)	
24–29	0 (0)	25 (6.2)	25 (6.1)	
30+	0 (0)	33 (8.2)	33 (8.1)	
Gender				0.423
Male	3 (33.3)	88 (22)	91 (22.2)	
Female	6 (66.7)	312 (78)	318 (77.8)	
Underlying disease				0.073
Yes	3 (33.3)	381 (95.2)	21 (5.1)	
No	6 (66.7)	19 (4.8)	388 (94.9)	
Program of study				1
Not health-related	6 (66.7)	252 (63)	258 (63.1)	
Health-related	3 (33.3)	148 (37)	151 (36.9)	
Household income				0.041
<15,000	6 (66.7)	141 (35.2)	147 (35.9)	
15,000–55,000	1 (11.1)	201 (50.2)	202 (49.4)	
>55,000	2 (22.2)	58 (14.5)	60 (14.7)	
Location of hometown				0.51
Three southern border provinces	2 (22.2)	128 (32)	130 (31.8)	
Other southern	6 (66.7)	247 (61.8)	253 (61.9)	
Other parts of Thailand	1 (11.1)	25 (6.2)	26 (6.3)	
Religion				0.767
Buddhist	5 (55.6)	247 (61.8)	252 (61.6)	
Muslim	4 (44.4)	147 (36.8)	151 (36.9)	
Other	0 (0)	6 (1.5)	6 (1.5)	
History of COVID-19 infection				0.5
Yes	1 (11.1)	29 (7.2)	30 (7.3)	
No	8 (88.9)	371 (92.8)	379 (92.7)	
Report of friends/relatives previously infected with COVID-19				1
Yes	5 (55.6) 4 (44.4)	209 (52.2) 191 (47.8)	214 (52.3) 195 (47.7)	
No				
Likely to receive the COVID-19 vaccine if recommended to by their doctors or pharmacist				0.011
Yes	5 (55.6) 4 (44.4)	359 (89.8) 41 (10.2)	364 (89.0) 45 (11.0)	
No				
Perceptions of COVID-19′s severity				0.006
Not severe, low, and moderate	6 (66.7)	94 (23.5)	100 (24.4)	
High	1 (11.1)	208 (52)	209 (51.1)	
Very high	2 (22.2)	98 (24.5)	100 (24.5)	
Perceptions of the safety of COVID-19 vaccines				<0.001
Not safe, low	8 (88.9) 1 (11.1)	83 (20.8) 317 (79.2)	91 (22.2) 318 (77.8)	
Moderate, high, very high				
Perceptions of the effectiveness of COVID-19 vaccines				0.101
Not effective, low	4 (44.4) 5 (55.6)	83 (20.8) 317 (79.2)	87 (21.3) 322 (78.7)	
Moderate, high, very high				
Self-reported anxiety				1
Low, moderate	3 (33.3)	122 (30.5)	125 (30.6)	
High	6 (66.7)	278 (69.5)	284 (69.4)	
Level of optimism				1
Very low, low, moderate	2 (22.2)	103 (25.8)	105 (25.7)	
High	5 (55.6)	204 (51)	209 (51.1)	
Very high	2 (22.2)	93 (23.2)	95 (23.2)	

**Table 2 vaccines-11-01157-t002:** COVID-19 vaccination status among university students in Southern Thailand.

Unvaccinated N = 18 (4.4%)			Vaccinated N = 391 (95.6%)	
Willingness to Receive the COVID-19 Vaccine				
Undecided	No	Yes	1 dose	2 doses or more
8 (1.96%)	1 (0.24%)	9 (2.2%)	30 (7.3%)	361 (88.3%)

**Table 3 vaccines-11-01157-t003:** Reasons for vaccine acceptance among university students in Southern Thailand.

Reasons for Vaccine Acceptation	N = 400 (%)
I am at risk of COVID-19 infection	112 (28.0)
I do not want to catch COVID-19	292 (73.0)
I do not want to spread COVID-19 to others	282 (70.5)
I trust in the efficacy of the COVID-19 vaccine	87 (21.8)
I am not concerned about the side effects of the COVID-19 vaccine	62 (15.5)
I want to be part of the fight against the pandemic	123 (30.8)
I want to return to normal life as soon as possible	357 (89.3)
Vaccination is free of charge	164 (41.0)

**Table 4 vaccines-11-01157-t004:** Reasons for vaccine hesitancy among university students in Southern Thailand.

Reasons for Vaccine Hesitancy	N = 9 (%)
I am not at risk of COVID-19 infection	2 (22.2)
I am not afraid of catching COVID-19	1 (11.1)
I want to wait until I have more experience with the new vaccine	4 (44.4)
I do not trust the efficacy of the COVID-19 vaccine	8 (88.9)
I am worried it will cause mild side effects (e.g., fever and pain at the injection site)	4 (44.4)
I am worried it will cause serious side effects (e.g., hospitalization and severe illness)	7 (77.8)
I do not trust public authority	7 (77.8)
I do not trust pharmaceutical companies	4 (44.4)
The media (e.g., TV and radio) has dissuaded me from getting vaccinated	2 (22.2)

**Table 5 vaccines-11-01157-t005:** Factors associated with the COVID-19 vaccine hesitancy among university students in Southern Thailand.

Variables	Crude OR (95% CI)	Adjusted OR (95% CI)	*p*-Value
Gender			0.342
Female	Reference	Reference	
Male	1.77 (0.43, 7.23)	2.55 (0.37, 17.62)	
Underlying disease			0.23
No	Reference	Reference	
Yes	5.73 (1.11, 29.46)	4.24 (0.4, 44.88)	
Program of study			0.867
Not health-related	Reference	Reference	
Health-related	0.85 (0.21, 3.45)	1.17 (0.19, 7.1)	
Household income			
>55,000	Reference	Reference	
15,000–55,000	0.14 (0.01, 1.62)	0.23 (0.01, 4.53)	0.336
<15,000	1.23 (0.24, 6.29)	1.4 (0.14, 13.93)	0.775
Likely to receive the COVID-19 vaccine if recommended to by their doctors or pharmacist			0.048
No	Reference 0.14 (0.04, 0.55)	Reference 0.16 (0.03,0.99)	
Yes			
History of COVID-19 infection			0.992
No	Reference	Reference	
Yes	1.6 (0.19, 13.23)	1.01 (0.06, 17.44)	
Report of friends/relatives previously infected with COVID-19			0.577
Yes	Reference 0.88 (0.23, 3.31)	Reference 1.64 (0.29, 9.33)	
No			
Perceptions of the safety of COVID-19 vaccines			0.003
Moderate, high, very high	Reference 30.55 (3.77, 247.72)	Reference 43.94 (3.62, 533.57)	
Not safe and low			
Perceptions of COVID-19′s severity			
Very high	Reference	Reference	
High	0.24 (0.02, 2.63)	0.45 (0.03, 7.06)	0.572
Not severe, low, and moderate	3.13 (0.62, 15.89)	4.32 (0.57, 32.68)	0.156
Perceptions of the effectiveness of COVID-19 vaccines			0.161
Not effective, low	Reference 0.33 (0.09, 1.25)	Reference 3.57 (0.6, 21.16)	
Moderate, high, very high			
Self-reported anxiety			0.782
Low, moderate	Reference	Reference	
High	0.88 (0.22, 3.57)	1.28 (0.23, 7.15)	
Level of optimism			
Very low, low, moderate	Reference	Reference	
High	1.26 (0.24, 6.62)	1.33 (0.14, 12.33)	0.802
Very high	1.11 (0.15, 8.02)	2.97 (0.24, 37.11)	0.399

## Data Availability

Data sharing not applicable.

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
