# Peer review of "Anxiety, Optimism, and COVID-19 Vaccine Hesitancy among Students in a University in Southern Thailand during the 2021 Academic Year"

_vaccines, 2023, doi:10.3390/vaccines11071157_

Round 1

Reviewer 1 Report

This is a good vaccine acceptance study on young people awareness in COVID-19 disease. The article is well written and with easy comprehensive reading. There are some doubts for me in sample numbers and quality. The sample is 409 in demographic table, but in methods they had only 375 students. If they joined the pilot group sample, the sample goes to 405 students, with 4 form undescribed. This aspect must be cleared, because join pilot and sample groups must be carefully performed, due to the quick evolution of pandemics. A month difference in sample collection is critycal and the isolated whole sample of 375 must be analysed isolated, as it was obtained at the same period, different from the pilot sample sample. obtained earlier in the  pandemics and therefore it could be differente from the whole sample.

Another aspects in the sample is the gender of respondents, mostly women.  Why the authors opt not to show gender effects in their sample? There are no differences in the proportions but it is important to present this absence in the other tables. Gender effect are usually present in anxiety evaluations. 

I believe that the those modifications could be easily performed by the authors, as they are minor modifications in manuscript structure.

Reviewer 2 Report

The authors performed a cross-sectional survey, from November and December 2021, to investigate the Covid-19 vaccine uptake in a Thai university. Overall, the manuscript is out of date, as the survey was performed one and half a years ago and WHO declared that the Covid-19 is no longer Public Health Emergency of International Concern (PHEIC) since May 4, 2023, although Covid-19 may exist permanently. Moreover, the results that “both anxiety and optimism regarding the COVID-19 pandemic were not associated with vaccine hesitancy” are contradictory, and the conclusion that “healthcare workers play an important role in COVID-19 vaccination counseling to prevent transmission” is not supported by the data in this manuscript, and such a conclusion is a common sense.

The study subjects were from a university only and it is unknown what the proportion of participants in all students in that university was. The participants appeared to be not representative for university students in southern Thai.

The data presented were difficult to follow. For examples, the authors mentioned “the sample comprised 375 university students” in the Method, but “This study comprised 409 students” in the Results. In Table 1, 400 students accepted the vaccination, but in Table 2, just 391 students were vaccinated. The description such “high levels of optimism regarding the pandemic” and “high levels of anxiety concerning the COVID-19 infection” is ambiguous.

In Table 1, only 9 students expressed the COVID-19 Vaccine Hesitancy. The percentage calculated by 9 students is easily biased. Similarly, the percentage calculated by 18 unvaccinated students in Table 2 is also easily biased. Such a small number cannot be used to calculate percentages.

None

Reviewer 3 Report

Thank you for the invitation to review this manuscript. This study investigates the outcomes that have been well discussed during the COVID-19 pandemic. I am not sure about the rationale and originality of this study. Why is this study needed? What research gap is covered by this study? these questions are not well described in this manuscript.

The method section is very briefly described. There is a need to provide several headings of the methods. The validation of questionnaire is missing. The authors need to provide detailed questions used to measure the outcome. It is not clear how the anxiety was measured? there are several valid tools to measure the anxiety such as GAD-7, but the authors opted other tool that is very less frequently used in the literature. 

Most of the findings in this study are already established in the literature. The discussion section should be without headings and should compare the findings first with national data from Thailand, followed by Asian data and finally with international or global data.

The criteria to defined vaccine hesitancy, vaccine acceptance and anxiety are quite vague, that may over- or under-represent the proportions of students who were vaccine hesitant, or acceptor. I am afraid whether the authors could address these issues in the manuscript.

Appropriate

Reviewer 4 Report

The study provides a valuable exploration into the prevalence of COVID-19 vaccine hesitancy among Thai university students. Using an online cross-sectional design, the authors effectively captured attitudes during a specific period and highlighted the significant role of healthcare professionals in vaccine counseling. With its focus on a critical demographic in the fight against COVID-19, your research contributes vital insights to the broader dialogue on vaccine hesitancy while informing potential health education strategies.

Upon reviewing your paper, I recommend performing additional analysis to bolster the robustness of your findings. Please follow the steps outlined below:

1)      Discriminant Power (Correlations): Calculate the discriminant power of each item within your scales. This involves computing each item's corrected correlation with the scale's total score, excluding the contribution of the item in question. Any item with a correlation value below 0.20 should be deemed as having low discriminant power.

2)      Discriminant Power (U mann with) High-Low Score Comparison: Compute the quartiles of the scale scores. Once you have these quartiles, compare the scores from the first quartile (those with the lowest scores) with those from the fourth quartile (those with the highest scores) for each item. To evaluate whether these two groups have a significant difference, employ a nonparametric test, specifically the Mann-Whitney U test.

3)      Construct Validity and Factor Analysis: Establish the construct validity of your scales by conducting a factor analysis. This analysis should reveal whether the scores on your scale tend to group into distinct factors. Only select characteristics with eigenvalues greater than one that can account for at least 3.5% of the variance. Then, allocate the items to each element considering the saturation of the thing with the factor; each article should have a saturation of at least 0.40 with its assigned factor. Please ensure each item is assigned to only one factor - the one with which it has the greatest saturation.

4)      In addition to the previous statistical analyses, I would also like to suggest creating a Directed Acyclic Graph (DAG) to examine the associations among different variables.

A DAG is a statistical tool widely used in epidemiology and other research fields for visualizing and understanding the relationships between multiple variables. It is particularly useful for identifying potential confounding factors or selection biases in your study. In essence, it aids in establishing the causal relationship between your variables and helps in understanding the complex interdependencies that may exist.

Using a DAG will allow you to better illustrate the pathways by which your variables might be related and can help in the construction of more accurate statistical models. It also assists in ensuring that the conclusions drawn from your study are robust and reliable.

To create a DAG, you can use a free program called Dagitty, available online at dagitty.net. The program is intuitive and user-friendly, and it can help you construct and analyze DAGs in a straightforward manner.

For guidance on how to use Dagitty and how to construct a DAG, you can refer to this example: https://www.ncbi.nlm.nih.gov/pmc/articles/PMC8267856/

Round 2

Reviewer 2 Report

As I mentioned in the first review, this cross-sectional survey on the Covid-19 vaccine uptake in students from a Thai university is out of date and the limited number of study participant from a university cannot represent the students in Thailand. Thus, the revision cannot alter these fundamental flaws.

The data presentation is confusing. In the Methods, the author mentioned “The sample size included 375 students; …”. But in the results, the authors stated “This study comprised 409 students”. The size of 375 students might be a calculated number of participants required in this study. The authors stated “The sample size included 375 students; based on Cochran’s formula, with a confidence level of 95% and error margin of 5%”, but it is impossible to calculate the required number of study subjects without a putative vaccine acceptance rate. Moreover, the subjects from a university cannot represent the overall situations in southern Thailand. Thus, the study design has fundamental flaws  

Table 4 shows that N = 400 students. Is this correct?

Overall, this study only reported the Covid-19 vaccination situations in 409 students in a university. The conclusion is not scientifically sound. 

None

Reviewer 3 Report

Thank you for revising the manuscript. The authors did not consider some suggestions in the manuscript.

There is a need to provide detailed information in the methods section. Please follow the headings Ethics, Study location, Study Design, Study Population (inclusion and exclusion criteria), Sample size estimation, sampling technique, study instrument (questionnaire) - [this heading will include development, translation, reliability and validation process], components to study instrument [this heading will have information about the number of items in questionnaire, divided into various sections, this section should also discuss that how the items were scored), Study outcomes, Statistics.

I do not think that without clear and comprehensive information on the methods, one can assess the validity of the findings.

Discussion: The authors have added the first passage during the revision and claimed that this is the first study of its kind having younger population etc. I would like to suggest to the authors to include objectives that have been assessed in the younger population in this study. The second passage can be merged with the first passage. 

There is a need to provide the reason for the low response rate. Please provide the risks of convenience sampling and cross-sectional study design. 

The violent situations discussed in the response letter have not been discussed in the discussion of this manuscript.

Please provide some information about the current vaccination coverage in Thailand (one dose, two doses and booster doses) as per official figures from WHO or health authorities in thailand.

Ok

Round 3

Reviewer 2 Report

As I mentioned in the previous review, the revision cannot change the fundamental flaws in this manuscript.

Reviewer 3 Report

Thank you for revising the manuscript. Please provide vaccination coverage in Thailand in the introduction section before problem statement. The current status of vaccination in the country should be provided before discussing any issue in the country.

ok
